# Detection and Localization of Albas Velvet Goats Based on YOLOv4

**DOI:** 10.3390/ani13203242

**Published:** 2023-10-18

**Authors:** Ying Guo, Xihao Wang, Mingjuan Han, Jile Xin, Yun Hou, Zhuo Gong, Liang Wang, Daoerji Fan, Lianjie Feng, Ding Han

**Affiliations:** 1School of Information Engineering, Inner Mongolia University of Science and Technology, Baotou 014010, China; gy_imu@163.com; 2College of Mechanical and Electrical Engineering, Inner Mongolia Agricultural University, Hohhot 010018, China; 3College of Electronic Information Engineering, Inner Mongolia University, Hohhot 010021, China; 31956032@mail.imu.edu.cn (X.W.); 32156014@mail.imu.edu.cn (M.H.); 32156123@mail.imu.edu.cn (J.X.); 32256011@mail.imu.edu.cn (Y.H.); 32356022@mail.imu.edu.cn (Z.G.); wangliang-1981@163.com (L.W.); fandaoerji@imu.edu.cn (D.F.); feng@haokuai.cn (L.F.); 4Inner Mongolia State Key Laboratory of Reproductive Regulation and Breeding of Grassland Livestock, Hohhot 010020, China

**Keywords:** target detection, objective positioning, coordinate regression

## Abstract

**Simple Summary:**

We proposed a target detection algorithm based on the channel attention mechanism SENet, the GeLU activation function and layer normalized ShallowSE. We refined and simplified the PANet part and the YOLO Head part in YOLOv4 to obtain the Custom_YOLO target detection module. We designed a 3D coordinate regression algorithm for three fully connected networks in order to predict the goats’ coordinates. We combined the improved YOLOv4 target detection algorithm and coordinate regression algorithm to achieve goat localization.

**Abstract:**

In order to achieve goat localization to help prevent goats from wandering, we proposed an efficient target localization method based on machine vision. Albas velvet goats from a farm in Ertok Banner, Ordos City, Inner Mongolia Autonomous Region, China, were the main objects of study. First, we proposed detecting the goats using a shallow convolutional neural network, ShallowSE, with the channel attention mechanism SENet, the GeLU activation function and layer normalization. Second, we designed three fully connected coordinate regression network models to predict the spatial coordinates of the goats. Finally, the target detection algorithm and the coordinate regression algorithm were combined to localize the flock. We experimentally confirmed the proposed method using our dataset. The proposed algorithm obtained a good detection accuracy and successful localization rate compared to other popular algorithms. The overall number of parameters in the target detection algorithm model was only 4.5 M. The average detection accuracy reached 95.89% and the detection time was only 8.5 ms. The average localization error of the group localization algorithm was only 0.94 m and the localization time was 0.21 s. In conclusion, the method achieved fast and accurate localization, which helped to rationalize the use of grassland resources and to promote the sustainable development of rangelands.

## 1. Introduction

In order to strengthen the scientific management of grazing and reduce human consumption, we should use information technology to manage pastures. The detection and localization of goats in natural grazing conditions are important in range management. On the one hand, detecting and locating goats prevents them from being lost and reduces losses for ranchers. On the other hand, detecting and locating goats can be used to track the foraging paths of goats as a way to study the correlation between goat movement and fleece, milk and meat production. Lastly, we can develop precise feeding programs for individual goats in order to improve the economic efficiency of farms [1]. Therefore, detection and localization studies for goats are of great value.

The current research focuses on the individual detection and individual localization of animals. Animal positioning is divided into indoor positioning and outdoor positioning. Relatively little research has been carried out on indoor positioning. Asikainen M et al. proposed a specific metric vector iterative positioning algorithm that took into account the limitations of WSN nodes in terms of computational power and energy usage, and achieved a positioning accuracy of 3 m with a lower communication overhead [2]. M. Hierold et al. implemented the encounter detection and simultaneous localization of bats in their natural habitat with a low-weight wireless sensor node [3]. N. Duda proposed an enhanced version of the developed mobile node, which had a range of up to 420 h in the BATS project for small-sized animals like bats [4]. L. Catarinucci used a novel RFID-based approach that enabled the effective localization and tracking of small-sized laboratory animals, and tests were conducted on rats to prove the validity of the method [5]. H. Millner proposed a system for the 3D indoor and outdoor localization of animals using a sequential Monte Carlo method by incorporating the dynamics of the target object [6]. The core part of outdoor positioning is the use of satellite positioning systems to determine the latitude and longitude information of livestock, which is then combined with other sensors and wireless communication technologies to achieve a wider range of applications [7]. In recent years, LoRa was used as a means of localization to measure the location of target nodes using algorithms, such as the time difference of arrival, time of arrival and indication of received signal strength [8,9]. P. Singh et al. proposed a system based on the Robust Principal Component Analysis (Robust PCA) that spatially localizes the animals in the image, and this system was better than a pre-trained R^3^Net [10].

As far as animal detection is concerned, with the booming development of artificial intelligence, more detection algorithms are being applied to livestock [11,12,13,14,15]. Target detection is mainly divided into two categories. The first consists of anchor-based target detection algorithms, which use a priori box-assisted models for prediction. The current two-stage R-CNN family of algorithms [16,17] and the single-stage SSD algorithms [18] are anchor-point-based algorithms. Another class of anchorless target detection algorithms that use the key points or the centroid of the target object for prediction are represented by YOLOv1 [19], Corner-Net [20] and CenterNet [21]. Dong W achieved 92.49% accuracy with his constructed dairy goat dataset using the faster R-CNN [22]. Ran Y implemented target detection for pigs using MobileNetv2-YOLOv3, with an accuracy rate above 97% [23]. Lei, J et al. proposed an improved detection method, termed YOLOv7-waterbird, by adding an extra prediction head, a SimAM attention module and a sequential frame to YOLOv7, and achieved a mean average precision value of 67.3%, enabling real-time video surveillance devices to identify attention regions and perform waterbird monitoring tasks [24]. Shuang Song et al. used a pruning-based YOLOv3 deep learning algorithm to detect sheep feces, obtaining a mAP value of 96.84% [25]. Taejun Lee et al. identified individual cattle using a YOLOv8-based model, with an accuracy of 97% [26]. Yu Zhang et al. proposed an integrated goat head detection and automatic counting method based on YOLOv5, with a detection accuracy of 92.19% [27].

Given that none of the above studies incorporated target detection and localization, the approach proposed in this paper is more suitable for the smart management of rangelands. This paper proposes a goat detection and localization method based on machine vision. The innovations of this paper are as follows:(1)We construct a goat image dataset. The dataset has a total of 8387 goats and consists of 11 goats in their natural grazing state.(2)We propose a goat target detection algorithm consisting of ShallowSE and an improved Custom_YOLO based on YOLOv4. We first added the attention mechanism to enhance the feature extraction capability of the module. We then changed the activation function to enhance the generalization of the model. Finally, we lightened the model to improve detection speed while ensuring accuracy.(3)We propose a 3D coordinate regression algorithm based on fully connected networks. We constructed a fully connected network to fit the transformation relationship between the 2D and 3D coordinates of goats. The network can calculate the spatial coordinates of a goat after detecting the goat.

## 2. Materials and Methods

### 2.1. Dataset Creation

The experimental site was located in a pasture of the Otokqi sub-field of the Inner Mongolia Yi-Wei White Velvet Goat Limited Liability Company and the experimental subjects were Albas velvet goats. The ranch covers an area of 32 square kilometers and we laid out 14 PTZ cameras, each with a field of vision of two kilometers. Image acquisition was performed using camera No. 12, as shown in Figure 1a. The acquisition period was from 10 to 12 June 2021. The camera was an IPC3526-X22R4-SI 2-megapixel starlight infrared dome network camera. The camera was connected to a computer that directly saved the video of the goats foraging outside. After frame separation and the removal of unclear images from the video, a total of 8387 images were obtained, of which 80% were used as a training set and 20% as a test set.

Positioning data were collected from GPS information via a Tibbo UG908 locator from Beijing, China with a positioning accuracy of 1.5 m, as shown in Figure 1b. The GPS data were collected on 12 June 2021. The Tibbo UG908 locator was strapped to the head goat. The data were sent at a frequency of 1 s/time and the computer received GPS data from the head goat in real time.

### 2.2. Experimental Platform

The target detection experiment used Python with the Pytorch deep learning framework, which was run on an Intel (R) Core (TM) i9-11900 K CPU with a NVIDIA RTX-3090-24 G graphics card and 64 GB RAM. This experiment did not use a pre-trained model. Every 16 images comprised a batch, there were 100 epochs of thawed training data and the initial learning rate was set to 10^−3^. The 3D coordinate regression experiment used Python with the Pytorch deep learning framework, as well as a NVIDIA GeForce GTX 1660Ti video card and 16 GB of memory. Each model was separately trained for 20,000 epochs, with an initial learning rate of 10^−4^, using GPU-accelerated training and Adam for network optimization.

### 2.3. Construction of the Goat Target Detection Algorithm

The goat target detection model proposed in this paper was based on YOLOv4 and the shallow convolutional neural network was first used as the backbone of the target detection model. Then, the YOLO Head and PANet (Position Attention Network) parts were streamlined.

#### 2.3.1. SE Module

In convolutional neural networks, each convolutional operation is performed for all channels within the convolutional kernel perceptual field. Although the spatial features of local regions can be fused with channel features, the feature differences between channels are ignored. Hu et al. proposed the channel attention mechanism SENet [28] to address these problems. The SE module can enhance the network’s ability to learn important features and improve learning efficiency. The basic structure of the SE module is shown in Figure 2 and contains four main stages: preprocessing, squeeze, excitation and reweight.

Before entering the SE module, a simple transformation of the input feature map, i.e., the mapping of *X* to *U* in the graph, was required. The procedure is shown in the following equation:$$F_{tr}:X\to U,X\in R^{H\prime \times W\prime \times C\prime },U\in R^{H\times W\times C}$$
where *H* is the height, *W* represents the width and *C* represents the number of channels.

After performing the normalized transformation, the feature map was globally averaged and pooled to compress the feature map into the global features, which was calculated as follows:$$z_{c}=F_{sq}\left(u_{c}\right)=\frac{1}{H\times W}\sum_{i=1}^{H}\sum_{j=1}^{W}u_{c}\left(i,j\right)$$
where *u_c_* ∈ *R^H^*^×*W*^, *z* ∈ *R^C^*.

After that, SENet applied the excitation operation to the global features to calculate the correlation between different channels. First, the global features were reduced to C/r dimensions by a fully connected layer and ReLU activation function; after that, the dimensions were recovered again by one full connection and Sigmoid activation to obtain a weight matrix composed of the weights of different channels, which was calculated as follows:$$s=F_{ex}(z,W)=\sigma (g(z,W))=\sigma (W_{2}\delta (W_{1}z))$$
where $W_{1}\in R^{\frac{C}{r}\times C}$, $W_{2}\in R^{\frac{C}{r}\times C}$ and *r* is the hyperparameter and the ReLU activation function.

Finally, the reweight operation multiplied each channel in the feature map with its corresponding weight to obtain a feature map with the weight information. The calculation formula is as follows:$${\tilde{x}}_{c}=F_{scale}(u_{c},s_{c})=u_{c}\times s_{c}$$
where $\tilde{X}=\left[{\tilde{x}}_{1},{\tilde{x}}_{2},\ldots ,{\tilde{x}}_{c}\right]$, $u_{c}\in R^{H\times W}.$

After going through the SENet module, the model was able to put more resources into the learning of channels with larger weights based on the different importance levels of channels, achieving the reinforcement of learning for important feature channels and placing a weaker emphasis on learning for non-important feature channels, which improved learning efficiency.

#### 2.3.2. GeLU Activation Function

In image processing tasks, the role of the activation function is to attach nonlinear properties to the neural network [29]. One of the most commonly used is the ReLU function, as shown in the curve in Figure 3. When the input is *x* > 0, the derivative of ReLU is constantly equal to 1, which can effectively solve the problem of gradient disappearance. However, when the input information is *x* ≤ 0, the derivative of the function is 0, which makes it extremely sensitive to abnormal inputs, resulting in the network being unable to back-propagate and causing the neurons to deactivate. So, we adopted GeLU activation, which can reduce the sensitivity of the activation function to outliers and enhance the generalization ability of the model.

The GeLU function is represented as:$$GeLU(x)=xP(X\leq x)=x\int_{-\infty }^{x}\frac{e^{-\frac{(X-\mu {)}^{2}}{2\sigma^{2}}}}{\sqrt{2\pi }\sigma }dX$$

#### 2.3.3. Layer Normalization

Normalization plays a crucial role in computer vision: by calculating the mean and variance, and then normalizing the input data to keep the scale of the data within a set range, the network can effectively avoid the problem of gradient disappearance and gradient explosion during training [30].

In previous image processing tasks, scholars generally used batch normalization, which normalizes all training samples in a mini-batch, as shown in Figure 4a where *C* denotes the number of channels, *N* denotes the batch size and *H*, *W* denotes the aspect of the feature map. Batch normalization is too dependent on the batch size, which will produce different means and variances for each batch, and when the batch sizes are inconsistent, the final processing results may not be representative of the overall data distribution.

To address this drawback of batch normalization, which has a strong correlation with batch size, Jimmy Lei Ba proposed layer normalization [31], which is processed as shown in Figure 4b. Layer normalization does not consider the size of each batch, and the mean and variance are computed for all neurons within the same layer, which is followed by normalization.

ConvNeXt mimics Transformer and Swin-Transformer by replacing batch normalization with an improved layer normalization, achieving an accuracy improvement rate of 1% when batch normalization is used [32]. As shown in Figure 4c, the layer normalization in ConvNeXt is a refinement of the traditional layer normalization, in which the setting of each pixel on all channels in the same layer of the feature map is normalized. ShallowSE borrowed the layer normalization design from ConvNeXt by replacing all instances of traditional batch normalization with this improved layer normalization method in the network structure.

#### 2.3.4. Model Streamlining

Considering that the dataset used in this paper is dominated by medium-sized targets, the number of a priori frames used for medium-sized feature layers is much higher than that for small and large sizes. To ensure that the accuracy rate does not decrease, we streamlined the YOLO Head, as shown in Figure 4, deleted the dashed part and kept the medium-sized feature layer. The part between the head and backbone is called the neck, and its main function is to fuse the semantic information contained in the effective feature map with the texture information, enhancing the expression ability of the feature map. Common necks include, among others, the FPN (Feature Pyramid Network) and PANet [33]. In this paper, we used the PANet as the neck section. Due to the single target class and the relatively small single target size in the dataset, we further streamlined the PANet, which reduced the original 5 convolutions to 3 convolutions. The red part in Figure 5 is the streamlined part and the streamlined target detection module is called Custom_YOLO.

### 2.4. Coordinate Regression Algorithm

Three coordinate regression models consisting of fully connected layers with ReLU activation functions were designed in this study. All three regression models used the ReLU activation function for nonlinear activation. SmallerFC used 2 fully connected layers, SmallFC used 3 fully connected layers and BigFC used 4 fully connected layers. The specific structures of the three networks are shown in Figure 6.

The dataset was organized according to the picture serial number, which is the sequence number of the picture when the video was split into frames. The coordinate conversion text dataset was obtained with a total of 5029 data points, a part of which is shown in Table 1. Among them, xmin, ymin, xmax, ymax, pan, tilt and zoom are input variables, and Xw and Zw are output variables.

Figure 7 shows the schematic diagram of the seven input variables. xmin, ymin, xmax and ymax represent the bounding box pixel coordinates of the target goats in the area in the image. Pan, tilt and zoom are the PTZ head parameters, which are the values in the lower left corner of the image. The output variables Xw and Zw are the values of the goat only in the direction of the Xw and Zw axes in the constructed world coordinate system. Figure 8 illustrates the flowchart of this study.

### 2.5. Model Training and Test Precision Evaluation

In order to evaluate the performance of the model, we evaluated the model in terms of mean average precision (mAP), loss, frames per second (FPS), parameters, multiply accumulate operations (MACs) and detection time as per the following metrics descriptions:

(1) mAP: The average accuracy of detecting all categories of targets, which is one of the important indicators for evaluating the performance of target detection models.

(2) Loss: Used to express the size of the deviation between the model’s prediction frame for the target and the target’s real frame. In the loss curve, the smoothness of the curve, the speed of convergence of the loss value and the size of the loss value after convergence are all references for judging the effectiveness of the model in fitting the dataset.

(3) MACs: Multiplying the cumulative number of operations, which can be interpreted as computational effort, is used to measure the time complexity of the algorithm/model.

(4) Parameters: Refers to the number of parameters in the network/model and measures the amount of computer storage space occupied by the model, which can be used as one of the indicators for determining the suitability of the network/model for mobile deployment.

(5) FPS: The number of images that can be processed in a second.

(6) Time: Time taken to detect an image.

## 3. Results

The YOLOv4 target detection module was used as a platform to test the performance of six backbone feature extraction networks with performance parameters including the number of parameters, computation, detection accuracy and detection speed. The test results are shown in Table 2.

The lightweight networks, MobileNetv3 and EfficientNet, were significantly smaller than the classical convolutional neural networks, VGG-16 and ResNet-50, in terms of number of parameters and computation time, but the lightweight networks were not necessarily faster than the classical network models in terms of inference speed. As can be seen from Table 2, the classical network was slightly better than the lightweight network in terms of detection speed when the difference in the average accuracy of the mAP was not significant. The proposed shallow convolutional neural network, ShallowSE, improved the FPS to be more than 20 frames per second, with guaranteed detection accuracy, which was a big improvement compared to the classical and lightweight networks. The mAP of ShallowSE was second only to the 96.34% of CSPDarkNet-53 and 95.53% of MobileNetv3.

### 3.1. Model Streamlining before and after Comparison

The models with CSPDarkNet-53, MobileNetv3 and ShallowSE as feature extraction networks were streamlined and noted as CSPDarkNet-53-Custom_YOLO, MobileNetv3-Custom_YOLO and ShallowSE-Custom_YOLO, respectively. We compared the original PANet and the simplified PANet results, as shown in Table 3.

As can be seen from Table 3, the total number of parameters in the streamlined model was further reduced. The CSPDarkNet-53-Custom_YOLO model had 51.60% fewer parameters than the original YOLOv4 model. The MobileNetv3-Custom_YOLO model had 77.21% fewer parameters than the original YOLOv4 model. The number of parameters in the ShallowSE-Custom_YOLO model was reduced by 87.75% compared to that in the original YOLOv4 model. The significant reduction in the number of parameters led to further improvement in the detection speed of the target detection model. The FPS of the CSPDarkNet-53-Custom_YOLO model was improved by 3.27 and the detection time was reduced by 3.3 ms when compared with the original YOLOv4 model. The FPS of the MobileNetv3-Custom_YOLO model was improved by 0.39 and the detection time was reduced by 1.6 ms when compared with the original YOLOv4 model. The FPS of the ShallowSE-Custom_YOLO model was improved by 3.38 and the detection time was reduced by 3 ms when compared with the original YOLOv4 model. The FPS of the Custom_YOLO model was improved by 3.38 and the detection time was reduced by 3 ms when compared to the original YOLOv4 model. In conclusion, compared with the model that had CSPDarkNet-53 as the main feature extraction network, the model designed in this paper showed a decrease in mAP, but also demonstrated a decrease in the number of parameters and detection time, and an improvement in FPS. The model we designed was generally better than the other models.

Figure 9 shows the comparison of the loss value curves before and after the streamlining of the three target detection models, where the blue curve is the loss curve before streamlining and the orange curve is the loss curve after streamlining. It can be seen that the optimized loss values converged slightly faster than the pre-optimized ones and the loss values after smoothing were smaller than those of the pre-optimized models, which indicates that the models fit the dataset faster and better.

### 3.2. Analysis of 3D Coordinate Regression Algorithm Results

The BigFC regression model was adopted in this paper and the experimental results were compared with the SmallerFC regression model and SmallFC regression model. The prediction results of the three regression models were compared using the training sets and test sets, and the results are shown in Figure 10 and Figure 11, respectively.

Figure 10 shows the prediction curves of the three networks using the training set compared with the true curves, where the blue curves are the true values of Xw and Zw, and the orange curves are the predicted values. In the comparison plot of Xw, SmallerFC and SmallFC fitted the training samples less well than BigFC. In 1~1000 samples, the prediction curve did not cover the true curve well. And starting from the 3500th sample, the goats moved rapidly in a short period of time causing the true curve to rise more. The difference between the prediction curve and the true curve was larger at this time. Relatively speaking, the prediction curve of BigFC basically covered the true curve and had the best fitting effect on Xw. In the comparison graph of Zw, the fitting degree of the first 1000 samples, SmallerFC and SmallFC were slightly inferior to BigFC. After that, the fitting degrees of the three were not much different. But the prediction curve of BigFC had significantly smaller up and down fluctuations, so the fitting effect was better. Overall, the BigFC regression model had a better fit on the training set.

Figure 11 shows the prediction curves of the three networks using the test set compared with the real curves. In the Xw prediction curve, the prediction effect of BigFC was significantly better than that of SmallerFC and SmallFC. And the fluctuation in the prediction curve for Zw was significantly smaller than that of SmallerFC and SmallFC. The prediction effect of BigFC was the best among the three.

Table 4 shows the prediction errors of the three regression models for the training and test sets. The average error of the model prediction decreased as the number of fully connected layers in the network increased. This indicates that the more the layers are fully connected, the better the model fitting and the smaller the prediction error. SmallFC increased by one fully connected layer compared to SmallerFC, which had a more significant reduction in the average error. BigFC worked best on both the training and test sets.

### 3.3. Goat Positioning Algorithm Results

Table 5 shows the statistical results of the localization errors of the three localization algorithms in 30 random samples. It was found that in the actual localization test, BigFC and SmallerFC had unsatisfactory localization results and performed less well than SmallFC in both maximum error and average error. The main reason was that SmallerFC only had two layers of full connectivity and underfitted the dataset, resulting in a higher maximum error and larger error fluctuations in the positioning accuracy test. BigFC had four layers of full connectivity and had excellent prediction curves in terms of both fluctuation and fit, which made it overfit to the dataset and meant it only predicted samples within the dataset well, but it did not have a good generalization ability for samples outside the dataset.

## 4. Discussion and Conclusions

This study proposed a vision-based goat localization method based on the existing gimbal of a sub-farm of Inner Mongolia Yiwei White Velvet Goat Co. This study proposed the shallow convolutional neural network, ShallowSE, obtained the Custom_YOLO target detection module after streamlining and optimizing YOLOv4, and constructed the goat target detection algorithm based on ShallowSE-Custom_YOLO. After experimental validation, the proposed goat target detection algorithm achieved a 95.89% mAP, 25.32 FPS and an 8.5 ms detection time with only 4.5 M parameters, which met the requirements of being lightweight and carrying out real-time and accurate target detection in practical application scenarios. In this paper, we proposed a 3D coordinate regression algorithm based on a fully connected network. Experiments showed that the fully connected network fit the dataset much better than the traditional machine-learning algorithm SVR and the more fully connected layers there were, the better the fitting effect. Finally, experiments were conducted on the vision-based flock localization algorithm proposed in this paper. The experimental results show that the localization algorithm using SmallFC with three fully connected layers as the coordinate regression model achieved the optimal localization accuracy and the localization speed decreased gradually with an increase in the number of fully connected layers.

On the one hand, the positioning algorithm for goats proposed in this paper expanded the functional utilization of the PTZ surveillance camera, replacing the wearable device positioning method. On the other hand, the video image-based localization method had better expandability and the localization method studied in this paper can incorporate more functions to better realize the intelligent management of pastures.

## Figures and Tables

**Figure 1 animals-13-03242-f001:**
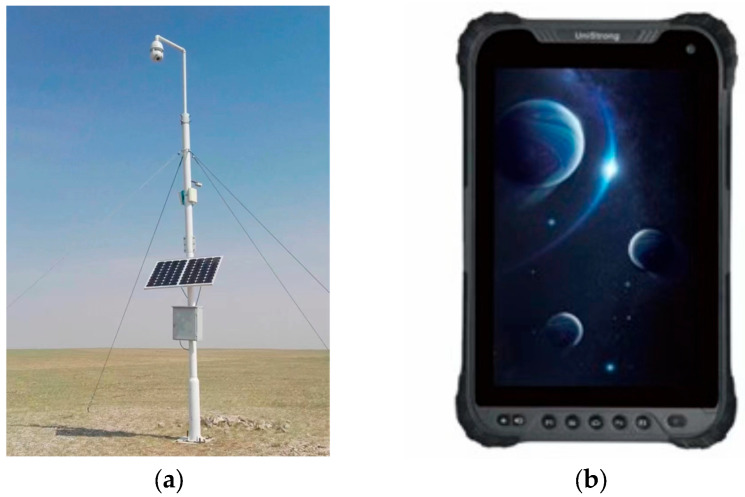
The picture (**a**) shows camera 12 at a height of about 15.6 m. The picture (**b**) shows the Tibbo GPS locator that was used to collect GPS data.

**Figure 2 animals-13-03242-f002:**
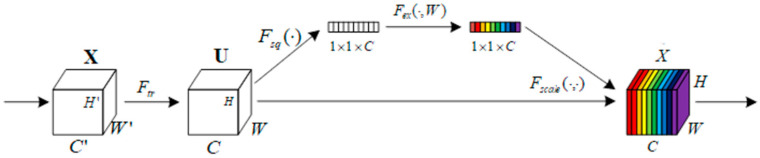
The structure of the SE module. *C* stands for number of channels, *H* stands for height and *W* stands for width. X→U is a mapping operation. $X\to \tilde{X}$ is the squeeze, excitation and reweight operation.

**Figure 3 animals-13-03242-f003:**
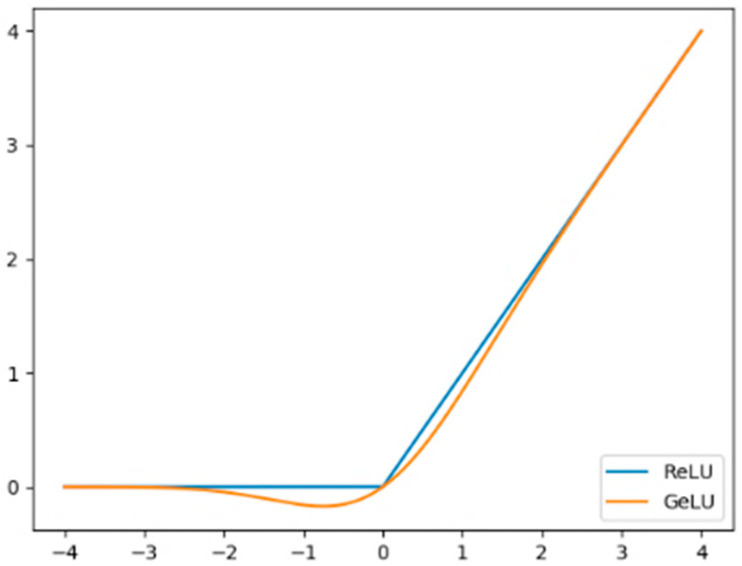
Comparison of GeLU and ReLU.

**Figure 4 animals-13-03242-f004:**
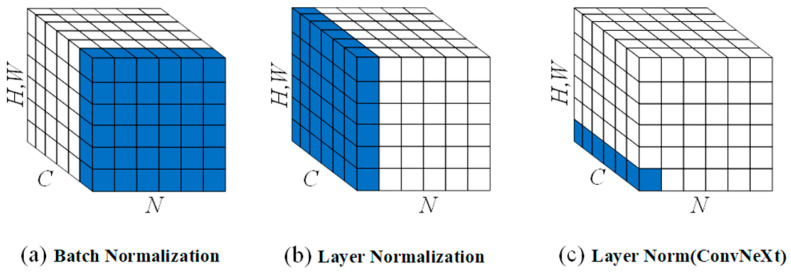
These are batch normalization method, layer normalization method and layer normalization method schematics, where *C* denotes the number of channels, *N* denotes the batch size and *H*,*W* denotes the aspect of the feature map.

**Figure 5 animals-13-03242-f005:**
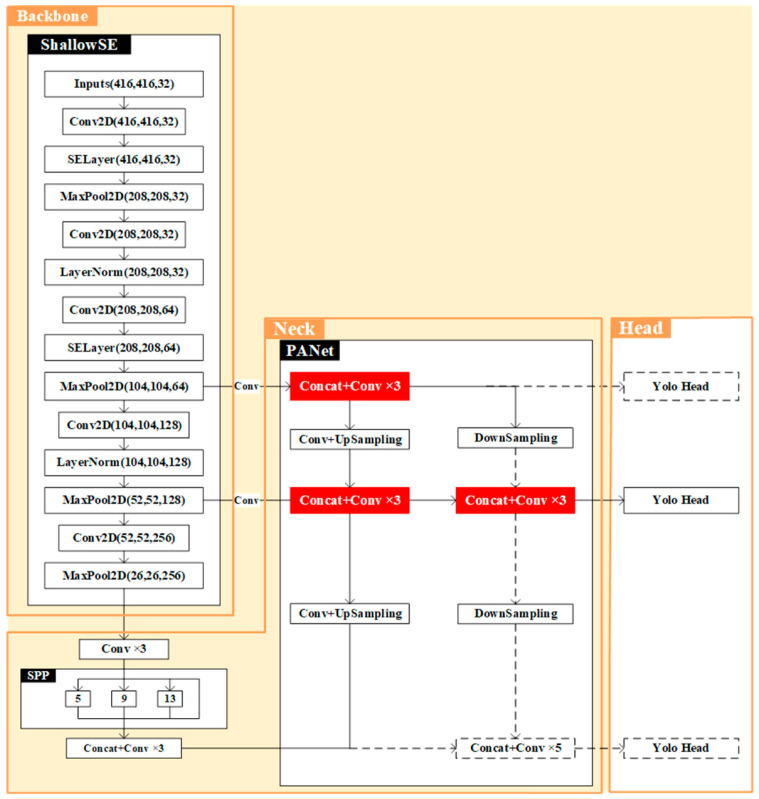
The structure of ShallowSE-Custom_YOLO.

**Figure 6 animals-13-03242-f006:**
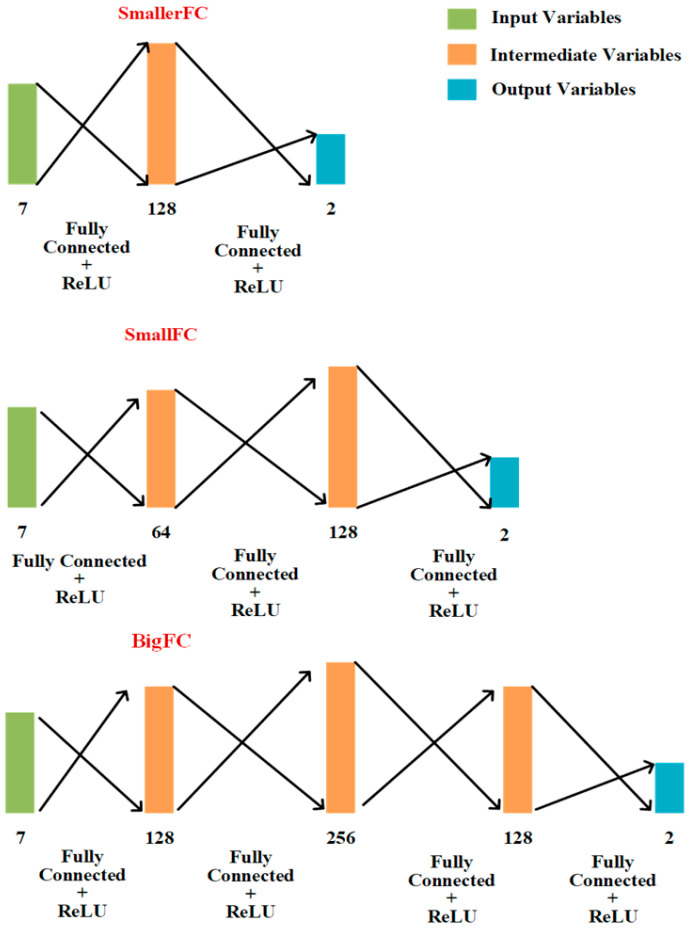
The structure of the coordinate regression model.

**Figure 7 animals-13-03242-f007:**
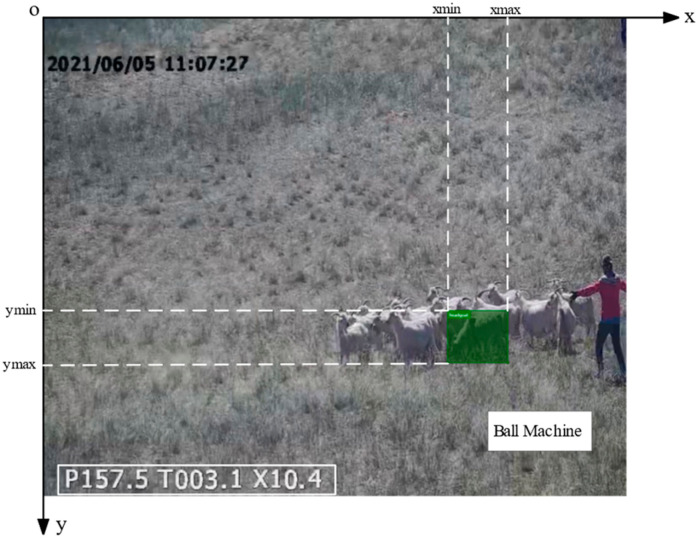
Diagram of input variables.

**Figure 8 animals-13-03242-f008:**
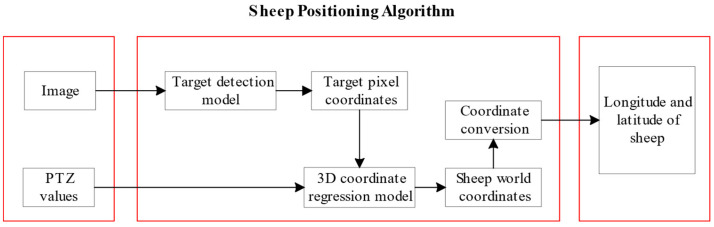
The overall structure of the goat positioning algorithm.

**Figure 9 animals-13-03242-f009:**
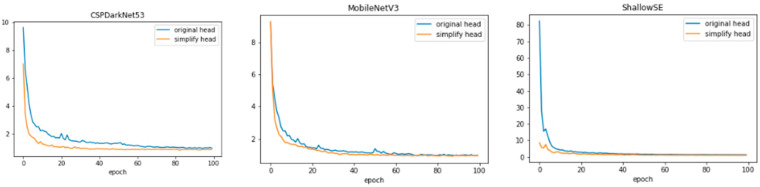
Comparison of loss function curves before and after model simplification.

**Figure 10 animals-13-03242-f010:**
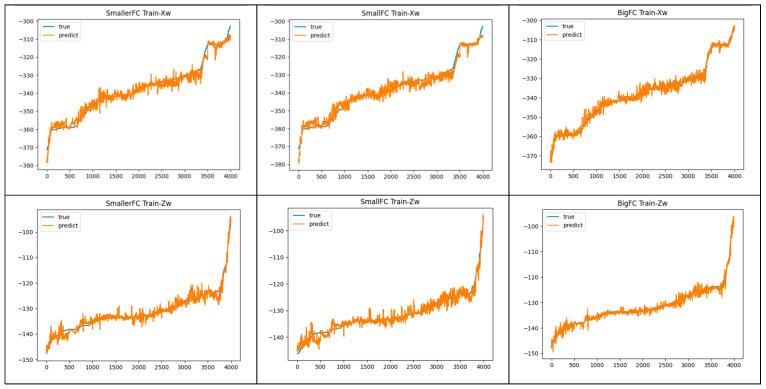
Prediction curve of the three fully connected networks using the training set.

**Figure 11 animals-13-03242-f011:**
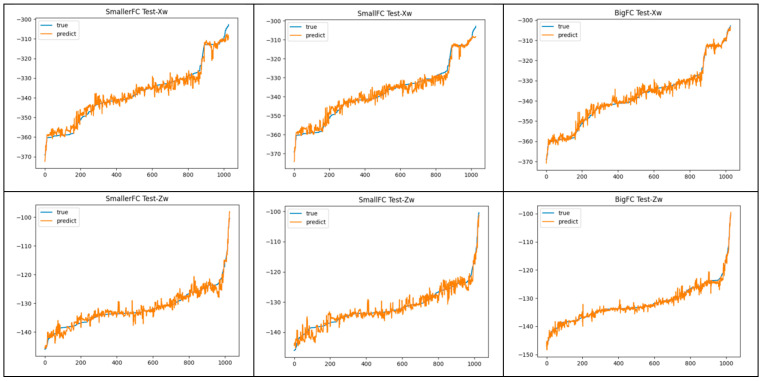
Prediction curve of the three fully connected networks using the test set.

**Table 1 animals-13-03242-t001:** Coordinate conversion text dataset.

Num	xmin	ymin	xmax	ymax	Pan	Tilt	Zoom	Xw	Zw
1	370.81399	322.28571	411.34543	400.20779	145.6	2.7	18.4	−302.75021	−136.18202
2	371.12577	321.66233	414.15146	403.32467	145.6	2.7	18.4	--	−136.18202
3	370.50221	325.40259	416.64570	403.94805	145.6	2.7	18.4	−302.75021	−136.18202
⋮	⋮	⋮	⋮	⋮	⋮	⋮	⋮	⋮	⋮
5028	341.64705	256.94117	365.17647	303.0588235	144.7	7.5	3.8	−371.25324	−97.453996

**Table 2 animals-13-03242-t002:** Test results of backbone feature networks.

Network	Body	Params (M)	MACs (G)	mAP	FPS
VGG-16	YOLOv4	14.714688	52.97694106	93.73%	19.19
ResNet-50	YOLOv4	23.508032	14.12160512	93.34%	18.30
CSPDarkNet-53	YOLOv4	26.617184	17.33899981	96.34%	13.75
MobileNetv3	YOLOv4	5.483032	0.76975596	95.53%	17.42
EfficientNet-B0	YOLOv4	3.595388	0.279041152	91.07%	16.64
ShallowSE	YOLOv4	0.400344	2.960644704	93.98%	21.94

**Table 3 animals-13-03242-t003:** Comparison before and after simplification of PANet.

Model	Params	mAP	FPS	Time (ms)
CSPDarkNet-53-YOLOv4	63,937,686	96.34%	13.75	29.1
CSPDarkNet-53-Custom_YOLO	30,943,462	96.18%	17.02	25.8
MobileNetv3-YOLOv4	42,231,118	95.53%	17.42	22.9
MobileNetv3-Custom_YOLO	9,623,966	96.09%	17.81	21.3
ShallowSE-YOLOv4	37,205,070	93.98%	21.94	11.5
ShallowSE-Custom_YOLO	4,556,458	95.89%	25.32	8.5

**Table 4 animals-13-03242-t004:** Prediction error of the three coordinate regression models.

Model	Training Set		Test Set	
	Average Error (m)	Maximum Error (m)	Average Error (m)	Maximum Error (m)
SmallerFC	1.3951541	8.1814829	1.37147070	8.7922006
SmallFC	1.0992516	8.3155968	1.05533216	8.3128948
BigFC	0.9719390	6.7282131	0.88486233	6.5844086

**Table 5 animals-13-03242-t005:** Statistics of positioning errors of the three algorithms.

Model	Longitude		Latitude	
	Maximum Error Distance (m)	Average Error Distance (m)	Maximum Error Distance (m)	Average Error Distance (m)
ShallowSE-Custom_YOLO-SmallerFC	4.3448	1.4355	5.5023	1.3414
ShallowSE-Custom_YOLO-SmallFC	2.0996	1.0433	3.3299	0.8292
ShallowSE-Custom_YOLO-BigFC	4.3176	1.9062	2.0578	0.8796

## Data Availability

Not applicable.

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
