# Peer review of "Detection and Localization of Albas Velvet Goats Based on YOLOv4"

_animals, 2023, doi:10.3390/ani13203242_

Round 1

Reviewer 1 Report

The authors presented original research work, with potential applications to livestock management. The work is quite interesting, but several aspects require attention.

Throughout the text, the words “sheep” and “goat” are used interchangeably. Being the research focused on Albas Velvet Goats, the vocabulary must be appropriately corrected.

Some references in the text are not reported according to the journal guidelines (e.g. Ba J L, 2016 - Liu Z, 2022 – Hendrycks D, 2016). Moreover, they are not reported in the reference list.

I suggest that the technical abbreviation terms present in the text (e.g. PANet) be explained at the first occurrence to improve the comprehensibility of the text.

Introduction

The introduction should be improved by adding references to relevant literature. In the first two paragraphs, more research can be cited to improve the background information on the effects of livestock management on pastures and on the purposes of studying animal positions in diverse livestock feeding systems. More appropriate citations than [3] and [8] can be cited to support the general statements.

Materials and methods

Is “1 second/time” correct?

In my view, in section 2.1 more detailed information about the context in which data were collected should be included (i.e. description of the pasture site, animals, managing conditions, how images were collected…).

It is not clear to me how location data are collected from each single goat using the GPS device.

In Figure 2, the meaning of all the letters must be explicated.

The description of Figure 4 is not clear, and some syntax errors must be fixed.

Results and Discussion

In Table 3, the explanation of the titles should be included in the body of the manuscript or as a footnote to the table.

In the comment to Table 4, the authors reported the same results presented in the table, the information is redundant.

No discussion is provided in this section.

Several typos and lexical/syntax inaccuracies are present in the manuscript, thus a general English revision to improve the quality of the presentation is necessary. 

Reviewer 2 Report

1Introduction: The manuscript proposes a machine vision-based approach for locating livestock in a ranch, which shows promising ideas. However, the introduction section needs further improvement. The overall structure is disorganized with excessive paragraphs. It should be summarized more effectively. Additionally, the manuscript lacks detailed descriptions in referencing, such as author identification, their contributions and conclusions. Furthermore, some paragraphs lack literature support.

2In the Materials and Methods section, the author describes the equipment used but fails to provide detailed explanations of the experimental design. Important parameters, such as the size of the experimental site, fixed position of the camera, and data collection time, should be addressed. As ranches have extensive areas, please ensure to enhance the description of your experimental design.

3Lack of model evaluation introduction, mAP? please expand the details.

4What exactly does "picture serial number" mean and how is it obtained? Please describe the data collection process in detail.

5We can understand that the author aims to obtain real coordinates through positioning collars.  However, please provide a detailed explanation of the specific method used to obtain these coordinates. Is the collar fixed on the body of the livestock, or is the positioning collar used to derive the pixel coordinates of all parts in the camera photos?

Round 2

Reviewer 1 Report

All the comments were answered and the manuscript was edited accordingly, thank you. The overall clarity of the work has now improved.

The paper still presents some inaccuracies (e.g. 1st line of the simple summary; 1st line of conclusions; reference [17] is not listed in the body of the manuscript, ...). I recommend carefully reviewing the general correctness of the manuscript.

Some minor English language adjustments are recommended.

Author Response

Comments 1: The paper still presents some inaccuracies (e.g. 1st line of the simple summary; 1st line of conclusions; reference [17] is not listed in the body of the manuscript, ...). I recommend carefully reviewing the general correctness of the manuscript.

Response 1: Thank you for reminding us of this important point. Now we have revised it according to the reviewer's comments. We have deleted the incorrected part. These changes can be found on the page 1, paragraph 1, line 1 and page 12, paragraph 2, line 1 to 2. We have rechecked the references to ensure that all of the documents in the references appear in the text. These changes can be found on the page 2, paragraph 3, line 5, page 5, paragraph 2, line 2 and paragraph 4, line 4, and page 6.

Comments 2: Some minor English language adjustments are recommended.
Response 2: We appreciate your reminder. We have touched up our manuscript using MDPI's touch-up service.

Reviewer 2 Report

It is suggested that the author further polish the article in English.

It is suggested that the author further polish the article in English.

Author Response

Comments 1: It is suggested that the author further polish the article in English.

Response 1: We appreciate your reminder. We have polish our manuscript using MDPI's touch-up service.